# Motion Capture Technology in Sports Scenarios: A Survey

**DOI:** 10.3390/s24092947

**Published:** 2024-05-06

**Authors:** Xiang Suo, Weidi Tang, Zhen Li

**Affiliations:** 1School of Athletic Performance, Shanghai University of Sport, Shanghai 200438, China; xiang_suo@sus.edu.cn; 2School of Exercise and Health, Shanghai University of Sport, Shanghai 200438, China; weidi_tang@sus.edu.cn

**Keywords:** motion capture, wearable sensors, visual recognition, pose estimation

## Abstract

Motion capture technology plays a crucial role in optimizing athletes’ skills, techniques, and strategies by providing detailed feedback on motion data. This article presents a comprehensive survey aimed at guiding researchers in selecting the most suitable motion capture technology for sports science investigations. By comparing and analyzing the characters and applications of different motion capture technologies in sports scenarios, it is observed that cinematography motion capture technology remains the gold standard in biomechanical analysis and continues to dominate sports research applications. Wearable sensor-based motion capture technology has gained significant traction in specialized areas such as winter sports, owing to its reliable system performance. Computer vision-based motion capture technology has made significant advancements in recognition accuracy and system reliability, enabling its application in various sports scenarios, from single-person technique analysis to multi-person tactical analysis. Moreover, the emerging field of multimodal motion capture technology, which harmonizes data from various sources with the integration of artificial intelligence, has proven to be a robust research method for complex scenarios. A comprehensive review of the literature from the past 10 years underscores the increasing significance of motion capture technology in sports, with a notable shift from laboratory research to practical training applications on sports fields. Future developments in this field should prioritize research and technological advancements that cater to practical sports scenarios, addressing challenges such as occlusion, outdoor capture, and real-time feedback.

## 1. Introduction

In the realm of sports, the pursuit of optimal performance, injury prevention, and effective training strategies has driven the continuous evolution of technology and scientific research [1,2]. Among the various tools and methods employed in sports science, motion capture technology has emerged as a crucial component in understanding, analyzing, and enhancing athletic performance. Motion capture refers to the process of recording and translating the movement of objects or people into digital data that can be analyzed and manipulated [3]. Its origins can be traced back to the 19th century when pioneering work by Eadweard Muybridge and H. C. Adam laid the foundation for capturing and studying human and animal locomotion [4].

Over the past few decades, significant advancements in software and hardware technology have propelled motion capture to new heights, expanding its applications across various fields, including rehabilitation, sports training, and human movement biomechanics [5,6]. In particular, the field of sports biomechanics, which integrates principles and methods from mechanics, anatomy, physiology, and other related disciplines to study the structure and function of the human movement system, has greatly benefited from the integration of motion capture technology [7]. By providing detailed kinematic and kinetic data, motion capture enables researchers and practitioners to gain deeper insights into the complex interplay of the nervous system, muscles, bone marrow, and joints during athletic performance.

One of the key areas where motion capture technology has made significant contributions is in postural analysis and balance assessment. Postural control and balance are essential factors that influence an athlete’s performance and susceptibility to injuries [8]. Traditional methods for assessing balance, such as force plates and laboratory-based motion capture systems, often face limitations in terms of cost, portability, and ecological validity. However, the advent of inertial sensors has revolutionized the field, offering a cost-effective and versatile solution for collecting and processing large amounts of athlete balance data in various settings [9].

The application of motion capture technology extends beyond postural analysis, encompassing a wide range of sports scenarios and research questions. From analyzing the biomechanics of swimming strokes [10] to investigating the kinematic factors influencing soccer kicking performance [11,12], motion capture has become an indispensable tool in sports science. Its ability to provide objective, quantifiable data has enabled researchers and coaches to optimize training programs, prevent injuries, and enhance overall athletic performance [2,13,14,15].

However, the rapid growth and diversification of motion capture technologies have also presented challenges for researchers and practitioners in selecting the most appropriate tools for their specific needs. Different motion capture systems, such as cinematography capture, electromagnetic capture, and computer vision capture, offer unique advantages and limitations that must be carefully considered [16]. Furthermore, the increasing complexity and volume of data generated by motion capture technologies necessitate the development of advanced analysis techniques, such as machine learning and pattern recognition, to fully harness their potential [17].

This review aims to provide a comprehensive overview of the various motion capture technologies used in sports scenarios, bridging the gap between technological advancements and practical implementation. By comparing and analyzing the characteristics, advantages, and limitations of each technology, we seek to guide researchers and practitioners in selecting the most suitable motion capture methods for their specific applications. Moreover, we discuss the current state-of-the-art applications of motion capture technology in sports, highlighting its potential to revolutionize athletic performance analysis, injury prevention, and rehabilitation.

## 2. Classification of Motion Capture Technology

To provide a comprehensive overview of motion capture technologies in sports scenarios, we conducted a thorough literature search focusing on the development, validation, and application of these technologies. The inclusion criteria for the selected articles were as follows: (1) the study involved the use of motion capture technology in sports contexts; (2) the article provided detailed information on the motion capture system, its validation, or its application; and (3) the study presented original research findings or technical advancements. Conference abstracts and studies focusing solely on non-sports applications of motion capture were excluded. The search results covered various motion capture systems, including traditional optical systems, wearable sensor-based systems, and computer vision-based approaches.

Among these technologies, traditional motion capture methods have long been considered the gold standard, typically relying on optical image analysis with markers as the benchmark. However, in recent years, the rapid development and application of deep learning have paved the way for wearable sensor-based motion capture and computer vision-based motion capture techniques [18]. Currently, motion capture technology can be broadly classified into two major categories: traditional optical systems [19] and computer vision-based motion capture technology, which has seen significant advancements in recent years. Additionally, wearable sensor technology and other types of technologies are also prominent in this field, as shown in Table 1.

### 2.1. Cinematography Capture Systems

Cinematography capture systems are generally considered the gold standard in the field of motion capture [34]. They utilize optical markers to calibrate the spatial coordinate system and accurately determine the real 3D coordinates of the subject being measured. The accuracy of optical triangulation in motion capture depends on several factors, including the relative positions of the cameras, the distance between the cameras and the object, the number and quality of optical markers, and the movement of the markers within the tracking space [35]. The trade-off between the sampling frequency and spatial resolution is also an important factor that affects the overall performance of the system. Typical application scenarios of modern optical motion capture technology are shown in Figure 1. The pipeline of cinematography motion capture involves the following key steps: synchronized multi-camera setup, marker placement on the subject, cameras capturing marker trajectory data, data processing and gap-filling, and mapping marker data onto a digital character to reconstruct high-fidelity 3D motion.

Cinematography capture systems can be divided into two types: active marker systems and passive marker systems. In passive marker systems, the markers used do not emit light, and the sensors rely on receiving reflected light from the markers for data recording. Representative products of passive marker systems include Vicon equipment. Compared to passive markers, active marker systems such as Optotrak3020 (Northern Digital, Waterloo, Canada) have stronger resistance to interference and greater stability [36]. However, the additional power supply and cables in active marker systems can interfere with the subject’s movement and actions [37]. Relying on fixed cameras for data collection, cinematography capture systems suffer from limitations such as a relatively fixed sampling area and a sampling range influenced by the number of cameras and their field of view. The maximum working range of optical sampling systems documented so far does not exceed 1000 m^2^ [38]. To meet the recording requirements of large-scale scenes, researchers have deployed over 20 cameras, significantly increasing the difficulty and cost of sampling. Furthermore, the calibration and synchronization efforts in post-processing also multiply. Another major bottleneck that is hard to overcome in image-based motion capture systems is occlusion. Some motion capture scenarios require restrictions on equipment placement, and the complexity of the subject’s movements leads to many gaps and blind spots in the collected data. This not only increases the workload of data processing but also reduces the precision of system output [39]. To meet the demands of testing in larger areas, an optical capture algorithm using a moving sampling camera has emerged. By mounting the camera on a sliding base that moves parallel to the direction of motion and using fixed markers on the ground for coordinate transformation, optical motion capture under linear motion conditions has been successfully achieved [40]. Although it comes at the cost of reduced accuracy and increased data processing time, this approach effectively reduces the complexity of motion capture systems for long-distance scenes. It has certain application value in motion capture scenarios such as short-distance sprints.

In addition, the iGPS system can achieve millimeter-level spatial positioning accuracy in a relatively large spatial scale [41]. However, such devices have limited performance in dynamic scenarios, as slightly higher speeds than walking can result in significantly increased positioning errors [42]. To meet the requirements of larger testing areas, an optical capture algorithm that uses moving cameras for tracking and sampling has been developed. By mounting the cameras on a sliding base parallel to the direction of motion and using fixed markers on the ground for coordinate transformation, optical motion capture under linear motion conditions has been successfully achieved [40]. Although this method sacrifices accuracy and increases the data processing time, it effectively reduces the complexity of motion capture systems for action acquisition at longer distances. This approach has potential applications in motion capture scenarios involving straight-line events such as sprints.

### 2.2. Electromagnetic Capture Systems

In addition to image capture systems, electromagnetic tracking systems are also an important component of motion capture technology. Wearable motion capture based on electromagnetic systems utilizes the calculation of the time difference between the reflection of electromagnetic waves from the base station to the target object to determine the target distance [43]. The main advantage of electromagnetic-based motion capture systems lies in their high tolerance for occlusion compared to optical motion capture systems. The positioning method, which does not rely on visible light, makes the impact of occlusion on such systems negligible [44]. A typical pipeline of wearable sensors in motion capture is shown in Figure 2. The wearable sensor pose estimation pipeline starts with calibrating the sensors to the body segments. Filtering is then applied to fuse the raw sensor data into cleaner orientation estimates. Linear accelerations are calculated by removing gravity, and double integrated to estimate segment position trajectories. Static periods are detected to apply zero-velocity updates, mitigating integration drift. Finally, a kinematic model combines the individual segment poses to reconstruct the full-body motion. 

There are IMU (Inertial Measurement Unit) wearable sensors, which typically integrate accelerometers and gyroscopes, and some products also integrate magnetometers. Wearable technologies based on IMU sensors have gained increasing prominence in the field of motion analysis due to their simple and reliable communication protocols, compact device size, and flexible application scenarios [45]. Additionally, there are UWB (Ultra-Wide Band) wearable sensors. Commercial UWB systems have demonstrated stable performance in tasks such as indoor motion tracking, system deployment, and sensor localization. They can achieve dynamic tracking in indoor sports scenarios with measurement errors below 0.5% of the working range, and the positioning data accuracy can be improved by adjusting the sampling frequency [46]. This technology holds greater potential in sports with larger motion ranges and higher speeds, such as the comprehensive speed and posture tracking of ski jumpers. In addition to IMU and UWB, the LPM (Local Position Measurement) system is another typical electromagnetic positioning system, which utilizes radar wave reflections between base stations and reference points for positioning [47]. LPM offers notable advantages in terms of the size of the positioning area, system deployment complexity, and post-processing of data [48]. In addition to IMU, UWB, and LPM systems, GNSS (Global Navigation Satellite System) wearable sensors utilize satellite signals for precise positioning and motion data. They offer advantages in sports with larger motion ranges, providing comprehensive speed and posture tracking. While they may face limitations in challenging environments, combining GNSS with other sensor technologies enhances accuracy [49].

### 2.3. Computer Vision Capture Systems

In recent years, the advancement of hardware computing power in parallel computing technology and the progress in deep learning have mutually benefited each other [50,51,52,53]. Deep learning-based computer vision techniques have been widely applied in specialized fields such as medicine, sports, and industrial monitoring [54,55,56]. Motion capture based on computer vision offers advantages such as higher accuracy, faster speed, and reduced workload [57,58].

Human Pose Estimation (HPE) plays a crucial role in sports scenarios, which involves analyzing motion action videos for single-person or multi-person to obtain two-dimensional or three-dimensional coordinate information about body poses, based on the number of synchronized input videos, and it can be categorized into monocular pose estimation and multi-view pose estimation. As shown in Figure 3, typical human pose estimation involves frame acquisition and preprocessing to prepare the input image, which is then passed through a convolutional backbone network and upsampling network to predict the locations of key body joints via heatmaps and silhouettes. The model architecture is selected based on requirements, and the chosen model is trained on a dataset using backpropagation. The trained model then undergoes inference, post-processing, evaluation, and fine-tuning to optimize its performance on localizing body joints in new images.

With the improved speed and reliability of algorithmic recognition, computer vision-based motion capture has started to be used in various fields, including rehabilitation training, sports performance enhancement, and biomechanical analysis [59,60]. Despite the advantages of higher accuracy and stronger robustness, there are limitations in the application of multi-view pose estimation algorithms, such as the inability to extract information from regular motion broadcast videos. Currently, in the field of motion and vision related to computer vision capture applications, most use pose estimation algorithms based on single-view visual recognition for 2D or 3D athlete pose estimation.

Firstly, in practical application scenarios, the motion recognition algorithm is the core of computer vision capture systems, and the dataset forms the foundation of algorithm training. Various visual recognition models trained based on datasets can convert human motion in video inputs into spatial coordinates or vectors. The subsequent analysis and inference system work based on the output of the visual recognition system for training and adjustment. In 2021, Xin Chen et al. released the SportsCap system, which achieved real-time 3D motion capture for most sports scenes. It significantly improved the accuracy of motion capture in similar task scenarios compared to traditional methods, and achieved satisfactory levels of action classification capability [61].

Secondly, due to the rapid relative motion between the target individuals and the fixed coordinate system in sports, severe motion blur is one of the obstacles hindering the further application of computer vision capture. To address this, researchers like Hong Guo developed PhyCoVIS, which combines motion capture technology and visualization techniques to transform athlete motion data into easily understandable and analyzable visual charts and images. Through the PhyCoVIS system, coaches and athletes can assess the athlete’s body coordination, identify potential issues, formulate corresponding training plans, and compare the coordination of different athletes and analyze the coordination of specific motion sequences [62].

The bottleneck of computer vision capture in sports and sports-related fields lies in the lack of specialized visual recognition datasets. In addition to the scarcity of datasets suitable for outdoor scenes, most of the datasets are general datasets established by computer science researchers for human daily actions, rather than specialized datasets for sports scenes [63]. Using general pose estimation datasets for computer vision capture training can lead to problems such as decreased recognition accuracy and difficulties in recognition. To address this, besides establishing specialized datasets for sports scenes, another possible solution is to utilize few-shot learning techniques. Traditional machine learning algorithms require large amounts of data to train models, but few-shot learning techniques can accomplish learning tasks with very limited data available [64].

Currently, computer vision capture has not yet produced widely recognized and widely used pose estimation algorithms for high-precision capture scenarios. However, in applications where there is a higher tolerance for motion localization errors, such as football tactical analysis research, computer vision capture technology has irreplaceable advantages such as non-interference, ease of deployment, and fewer limitations. Some mature specialized applications already have considerable value.

### 2.4. Other Motion Capture Systems

In 2018, Dawei Liang et al. successfully extracted contextual information from the audio portion of videos, surpassing the context information that can be obtained from image-based methods [30]. Similar systems that utilize deep learning-based audio classifiers provide a new dimension of information for distinguishing complex actions.

In addition to sound waves, motion capture techniques utilizing radar (Radar Modality) also hold significant practical value. Sevgi Z. Gurbuz et al. utilized continuous radar RF data for action recognition and employed Generative Adversarial Networks (GANs) for sample synthesis, achieving comparable accuracy to Vicon systems in terms of output precision [31]. Compared to video-based motion capture systems, radar-based motion capture not only offers the same advantages but also enables data collection while ensuring privacy protection. This unique technological advantage makes radar-based motion capture highly promising with great technical potential [65].

Beyond the range of motion capture methods that use specialized equipment, there is the Wi-Fi Modality, which utilizes the echoes of Wi-Fi signals for motion capture. By extracting spatial–temporal information from the Channel State Information (CSI), and using Convolutional Neural Networks (CNNs) as the model classifier, it becomes possible to achieve cross-scene motion capture and recognition [32]. The greatest advantage of this technology is the freedom from the constraints of dedicated devices for motion capture tasks. In an ideal scenario, it would be possible to estimate human motion within the working range using Wi-Fi signals. This holds significant importance for low-cost and lightweight research projects, as it provides an irreplaceable solution for motion capture without the need for specialized equipment.

Fusion Motion Capture (FMC), which combines multiple data acquisition techniques, is one feasible solution for motion recognition in sports scenarios. Timo Von Marcard et al. integrated visual motion capture with wearable IMU sensors to improve and stabilize whole-body motion capture techniques [66]. Brodie et al. used a similar multimodal motion capture system to extract key performance indicators related to athletes’ body and physiological limits, such as average drag coefficient and maximum tilt angle during turns, which help optimize game strategies [33]. Hui-Min Shen et al. proposed a Magnetic Measurement Unit (MMU) approach for multisensor fusion in motion capture, enhancing system output accuracy and stability [67]. Additionally, Hasegawa et al. developed an action-assisted monitoring system that utilizes sound wave information to monitor and provide feedback on the skier’s center of gravity, assisting skiers in overcoming instinctive reactions of leaning back [68]. This helps guide specific technical optimizations for athletes, improving performance and reducing the risk of injury. The SmartSki system, designed by Anton Kos et al., underwent functional testing and validation by several professional alpine skiers during a year of snowfield testing [69]. However, its lack of portability hindered its practical implementation, keeping the system mostly in the experimental phase.

The main advantage of multimodal motion capture technology lies in the use of different types of data acquisition methods to simultaneously measure the pose and motion information of the moving target in different environments, achieving precision and reliability beyond the reach of a single technology [70,71]. This also implies broader potential application scenarios, faster data analysis capabilities, and better data visualization potential. At the same time, multimodal motion capture technology requires relatively more complex system integration, larger data processing capabilities, more precise data monitoring capabilities, and higher hardware costs. Overall, multimodal human motion capture technology has significant advantages in motion capture and analysis, but it needs to overcome technical challenges such as high equipment costs, high complexity, and difficulty in algorithm design. In the future, with the continuous development of technology, multimodal techniques are expected to become important means for human motion analysis and virtual reality, bringing more convenience and assurance to human life and health.

## 3. Application of Motion Capture in the Field of Sports

Motion capture technology exhibits different forms of expression and usage scenarios depending on the research objectives and topics. Specific application areas include human kinematics, biomechanics, motion technique analysis, athletic performance, and more [72]. In particular, motion capture technology can be used for topics such as technical analysis, training, athletic assessment, and sports injury medicine for athletes. For example, analyzing a soccer player’s running routes, shooting angles, and power can provide coaches and athletes with reliable quantitative metrics to improve team tactics and skills [73]. In swimming competitions, motion capture technology can measure swimmers’ speed, angular velocity, and other technical motion kinematics information, enabling the analysis of swimming postures and fluid resistance effects to assist athletes in improving training efficiency [74,75]. This article focuses on the typical research application scenarios of motion capture technology in the field of sports; Figure 1 illustrates the priority indicators and typical application scenarios of different motion capture techniques.

### 3.1. Construction of Athlete Performance Datasets

Xiong Zhao et al. utilized the Raptor-E motion capture system to collect and analyze three-dimensional motion data from nearly 200 athletes, establishing a database of motion and performance [76]. This enables researchers and system users to utilize advanced analysis methods such as pattern recognition and machine learning to study athletes’ movement patterns and gain new insights into the potential correlations between movement patterns and injury history, sport specificity, competition level, and demographics. Raptor-E is a passive marker-based optical motion capture system that offers advantages such as high accuracy, real-time performance, and strong outdoor interference resistance [77]. When combined with dedicated visualization systems designed for specific sports, the management and application of athlete performance datasets become more reliable and convenient [78].

Optical-based motion capture systems perform better in projects with simpler limb movements. However, they have limitations in terms of a relatively fixed sampling domain, where the sampling range is influenced by the number of cameras and their field of view. The maximum working range typically does not exceed 1000 square meters [38]. Additionally, the systems face occlusion issues. The mainstream solution currently involves using multiple cameras for video signal capture, but this increases the workload of post-processing and reduces the accuracy of system output [39].

### 3.2. Real-Time Assistance for Athlete Training and Competition

Compared to optical-based methods, motion capture using wearable sensors imposes less interference on athletes. To obtain more natural human gait kinematic information, Julie Rekant et al. installed 24 wearable sensors on 10 healthy males while using the Vicon system as a ground truth reference [79]. The study found significant differences between the measurements obtained from the wearable sensors and the Vicon system across different subjects, with the largest errors occurring in the sagittal plane, where the mean angular error was close to 5°. This error was considered relatively noticeable.

Although wearable motion capture solutions based on electromagnetic systems have advantages such as strong interference resistance and simple device setup, the positioning principles of such solutions limit their performance in terms of angular measurement accuracy. Most of these solutions rely on time-of-flight methods for distance measurement and positioning, resulting in significant cumulative errors during continuous positioning operations. Compared to optical-based motion capture solutions, commercially available products currently achieve lower tracking and positioning accuracy. Therefore, in typical motion capture tasks, electromagnetic-based motion capture solutions are not the preferred choice. However, in some special scenarios, such as when the motion range is too large, lighting conditions in the environment are not ideal, or occlusion issues cannot be resolved, specific electromagnetic-based motion capture solutions can be considered, and the system’s positioning accuracy needs to be validated to meet the research requirements.

### 3.3. Multi-Camera Motion Capture Technology for Training

In 2019, Eduard Pons et al. conducted a validation study on the effectiveness of a multi-camera motion capture system. The research team tracked 38 official matches of male athletes in the Spanish second division league and collected thousands of valid data points. The team validated the statistical indicators of standardized mean deviation and coefficient of variation for the system data, demonstrating that multi-camera motion capture technology can meet the research and training requirements in sports scenarios, with high repeatability and stability [80]. The bottleneck of computer vision-based motion capture in sports and specific fields lies in the lack of dedicated visual recognition datasets [63,81]. Apart from establishing specialized datasets for sports scenarios [82,83], another possible solution is to utilize few-shot learning techniques, which enable training with a small amount of data [64].

Currently, there is no widely recognized and widely used pose estimation algorithm in computer vision-based motion capture technology for scenarios with high accuracy requirements. However, in applications where the tolerance for motion positioning errors is higher, such as football tactical analysis and other research topics, vision-based motion capture technology offers irreplaceable advantages such as no interference, easy deployment, and fewer limitations. Some mature specialized applications already have considerable practical value.

## 4. Discussion

In the context of sports, the application of motion capture technology in sports scenarios requires careful consideration of the following factors compared to technical development environments:(1)Indoor laboratory settings offer controlled environments with better control over factors like lighting, temperature, and humidity, reducing noise and interference. However, outdoor sports scenarios present complex and variable conditions, including wind, sunlight, shadows, and different surfaces (e.g., plastic, grass, snow, ice, water), which can impact the accuracy and stability of motion capture systems. Outdoor conditions also experience changes in temperature, humidity, and visibility, while non-isolated environments introduce sound, lighting, and electromagnetic interference, further affecting motion capture tasks. Overcoming these environmental limitations is crucial for ensuring accurate and reliable motion capture in sports.(2)System setup: Indoor laboratory settings offer easier installation and calibration of motion capture systems due to the controlled environment. However, outdoor sports scenarios present more complex conditions, requiring additional effort for system setup and adjustment to achieve higher accuracy and precision. The precision of the measurement method is inversely proportional to the effective working range of the system. In sports scenarios, where large-scale motion scenes are common, capturing human kinematic information requires additional software and hardware optimization methods. These challenges necessitate careful system setup and optimization to ensure accurate motion capture in outdoor sports environments [84].(3)Motion characteristics: The movements in indoor laboratory settings are usually simple and single, such as gait analysis or arm movements. For these types of movements, indoor motion capture systems can provide high-quality data. In outdoor sports scenarios, there may be more complex movements, such as aerial rotations or limb flips, which can be distorted or inaccurate due to environmental factors. Motion capture for sports scenarios often requires capturing fast movements. In specific research scenarios, such as ballistic analysis in shooting or instant analysis of baseball swings, the sampling frequency may exceed 1000 Hz [16].(4)Data processing: Data processing is relatively straightforward in indoor laboratory settings because the data in controlled environments are usually stable and accurate [85]. In outdoor sports scenarios, data processing needs to consider more factors, such as lighting and environmental noise issues, occlusion and penetration issues, motion model establishment, data noise, and filtering issues [86].

When selecting the appropriate motion capture technology for specific sports applications, it is crucial to comprehensively consider factors such as the size of the capture space, the complexity of the athlete’s movements, the real-time performance requirements of the system, and the acceptability of the equipment’s impact on the athlete’s movements.

Cinematography capture systems offer high accuracy and are considered the gold standard in motion capture. However, they face limitations in terms of fixed capture areas, occlusion issues, and the impact of markers on natural movements. These systems are more suitable for biomechanical analysis in controlled environments.

Electromagnetic capture systems, including IMUs, UWB, and LPM, provide advantages in terms of occlusion resistance, flexible scenarios, and real-time performance. IMU-based wearable sensors are compact and widely used in motion analysis, while UWB systems enable high-precision indoor tracking, and LPM offers benefits in positioning area and deployment complexity. However, electromagnetic systems may face limitations in accuracy compared to optical systems and require careful sensor placement.

Computer vision capture systems have made significant advancements with deep learning techniques. Pose estimation algorithms enable markerless motion capture from video inputs, offering high accuracy, fast speed, and reduced interference with natural movements. These systems have been applied in rehabilitation, sports performance analysis, and biomechanical research. However, computer vision systems currently lack large-scale sport-specific datasets, which limits their performance in complex sports scenarios. Few-shot learning techniques are a potential solution to address this limitation.

Multimodal motion capture systems, which fuse data from multiple sensors, provide a promising approach for complex sports scenarios. By combining the strengths of different capture technologies, such as vision and wearable sensors, multimodal systems can achieve high accuracy, robustness, and adaptability to various environments. However, multimodal systems also face challenges in terms of system complexity, data synchronization, and higher costs.

The application of motion capture technology in sports scenarios faces various challenges, including environmental limitations, system setup, motion characteristics, and data processing. In this paper, a comparison and evaluation of the application of motion capture technology across different sports scenarios were conducted, with primary consideration given to factors such as repeatability, robustness, accuracy, and reliability. Due to differences in application scenarios, testing standards, and testing conditions among various motion capture systems, the assessment of reliability and repeatability may not be directly comparable, even for the same motion capture method. To address this issue, this paper employed a normalized evaluation approach by analyzing the relevant literature covered. The performance metrics from different studies were converted into a unified evaluation criteria by comparing the reported values within each specific metric and assigning them to a normalized scale. In cases where explicit performance metrics were not provided in the reviewed literature, this paper made qualitative assessments based on the overall conclusions and discussions presented in those studies, aligning them with the normalized scale.

The normalization process involved identifying the best-performing motion capture technology within each performance metric category and assigning it a score of 100%. The scores for the remaining technologies were then calculated as a percentage of the best-performing technology’s score within that category. This approach allowed for a standardized comparison of performance metrics across different motion capture systems, despite variations in testing conditions and reporting methods. The normalized scale was then divided into four ranges: 0–40%, 40–60%, 60–80%, and 80–100%, corresponding to the terms “low”, “medium”, “medium to high”, and “high”, respectively. By comparing the normalized performance metrics of each motion capture technology against these percentage ranges, this paper was able to assign the appropriate qualitative descriptor to each technology’s performance in terms of repeatability, robustness, accuracy, and reliability. Table 2 summarizes the performance metrics of various motion capture technologies, offering preliminary references for technology selection in subsequent research and applications.

In summary, the choice of motion capture technology depends on the specific requirements and constraints of the sports application. Researchers and practitioners should carefully evaluate factors such as capture space, motion complexity, and equipment impact on athletes to select the most suitable motion capture technology for their sports analysis tasks. As motion capture technology continues to evolve, addressing the challenges posed by outdoor sports scenarios and leveraging advancements in deep learning and multimodal systems will be crucial for advancing the field of sports biomechanics and performance analysis.

## 5. Conclusions

Motion capture technology offers numerous advantages and applications in the field of sports, enabling the analysis of athletes’ technical performance, training effectiveness, competitive assessment, and sports medicine-related information. Computer vision-based motion capture technology exhibits high recognition accuracy and wide applicability in various sports scenarios, particularly in large-scale scenes like football matches. Wearable sensor-based motion capture technology has made significant advancements in accuracy and multifunctional monitoring, proving particularly effective in small-scale scenarios such as swimming competitions. Imaging-based motion recognition remains dominant, offering unparalleled accuracy advantages across most scenarios. As artificial intelligence continues to develop, motion capture technology in sports will become more extensive and profound, supporting healthier, more scientific, and efficient training methods.

The research outlook in this field focuses on the future development of motion capture technology in sports. With ongoing advancements in technology, motion capture will provide even more accurate and objective data analysis and optimization solutions for athletes and coaches. Computer vision-based motion capture will see broader applications as improvements in capture accuracy, real-time performance, and stability are achieved through the further development and optimization of computer vision technology. Wearable sensor-based motion capture will grow in areas such as health monitoring and sports training. Multimodal motion capture technology will become more accurate and comprehensive, serving as a reliable solution for capturing complex scene motions. Additionally, the integration of emerging technologies like artificial intelligence will optimize motion capture algorithms and models, enhancing the efficiency and accuracy of sports performance analysis. In conclusion, motion capture technology brings opportunities and challenges to the sports field, supporting the pursuit of healthier, more scientific, and efficient training methods.

## Figures and Tables

**Figure 1 sensors-24-02947-f001:**
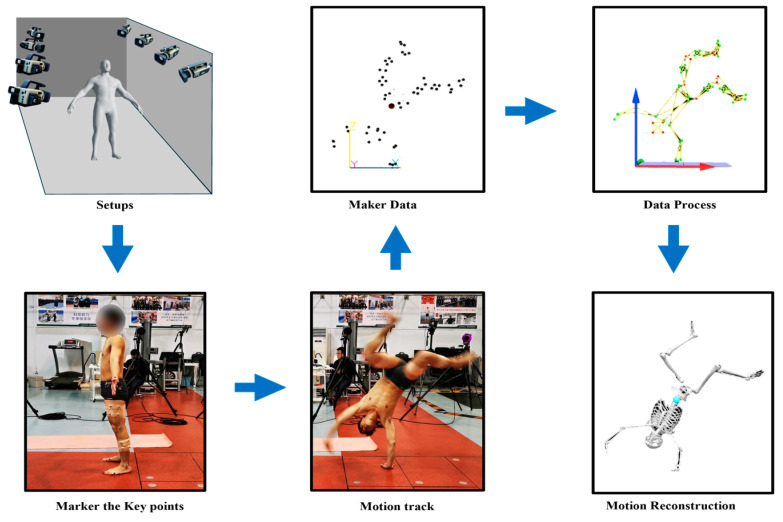
Pipeline of cinematography motion capture.

**Figure 2 sensors-24-02947-f002:**
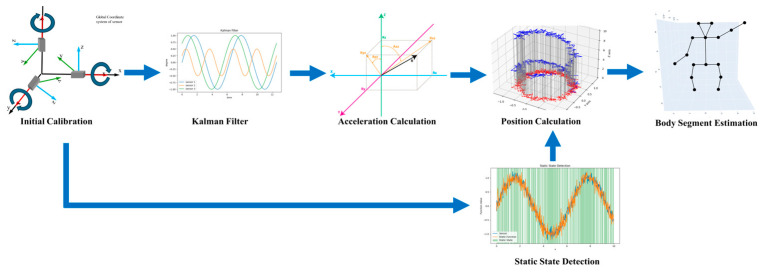
Pipeline of wearable sensor motion capture.

**Figure 3 sensors-24-02947-f003:**
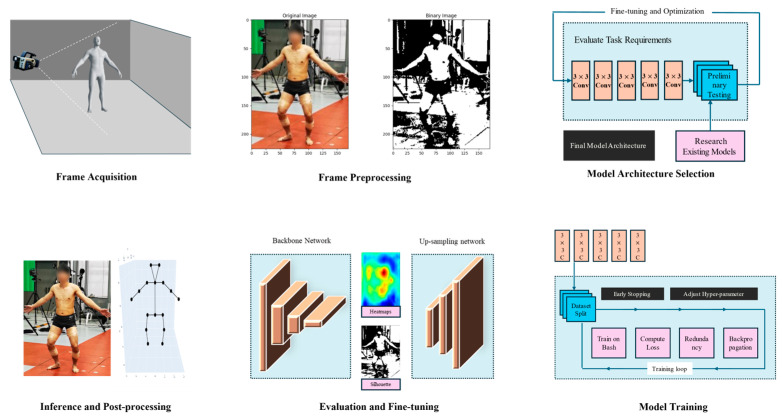
Typical pipeline of HPE.

**Table 1 sensors-24-02947-t001:** Classification of motion capture technology.

Category			Application Example
Cinematography Capture	Active Marker		Landing Technique [20]
	Passive Marker		Gait Analysis [21]
Electromagnetic Capture Systems	GNSS		Soccer Player Kinematic Data Acquisition [22]
	IMU		Motion Data Validation [23]
	UWB		Tennis Player Positioning [24]
	LPM		Youth Soccer Performance [25]
Computer Vision Capture	Single-Person	2D	Gait Analysis [26]
		3D	Handball Action Analysis [27]
	Multi-Person	Bottom-Up	Baseball Swing Assessment [28]
		Top-Down	Gait Analysis [29]
Other	Audio Modality		Activity Recognition [30]
	Radar Modality		Activity Recognition [31]
	Wi-Fi Modality		Cross-scene Action Recognition [32]
	Fusion Modality		Ski Racing Biomechanics [33]

**Table 2 sensors-24-02947-t002:** Comparative analysis of characteristic indicators of different motion capture methods.

Motion Capture Technology	Accuracy	Advantages	Constraints	Robustness	Repeatability	Reliability	Sports Scenarios	Sports Applications
Cinematography	High	High accuracy, suitable for complex movements	Limited capture volume, marker occlusion	Medium	High	High	Lab-based analysis, technique evaluation	Biomechanical analysis, technique optimization, injury prevention
Wearable Sensors	Medium	No marker occlusion, large capture volume, real-time tracking	Prone to electromagnetic interference, lower accuracy than optical systems	High	High	Medium to High	Indoor and outdoor training, competition monitoring	Real-time performance tracking, load monitoring, tactical analysis
Computer Vision	Medium to High	Markerless tracking, flexible setup	Line of sight, lighting, computationally intensive, sensitive to lighting conditions	Medium to High	High	Medium	Lab-based analysis, technique evaluation	Biomechanical analysis, technique optimization, movement pattern recognition
Others (e.g., Fusion Modality)	High	Integrating the advantages of multiple sensors	Sensor synchronization	High	High	High	Comprehensive performance analysis	Multifaceted performance assessment, injury risk prediction

## Data Availability

This article contains no data or material other than the articles used for the review and referenced.

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
