# Peer review of "Motion Capture Technology in Sports Scenarios: A Survey"

_sensors, 2024, doi:10.3390/s24092947_

Round 1
Reviewer 1 Report
Comments and Suggestions for Authors
The paper aims to supply a survey to guide researchers in selecting the most suitable motion capture technology for sports science investigations.
Weaknesses:
The Classification of Motion Capture Technologies section describes various capture technology systems. These are not commented on in the Discussion, which focuses on the differences between indoor and outdoor applications of motion capture technologies.
The abstract states, "Integrating AI technology with various motion capture methods provide unique capabilities not found in traditional approaches," but the article does not explain why this statement is true and what the term "unique capabilities" means.
Fig. 5 (line 380) is mysterious – values are unclear and were not commented in the text. What was the source of these data?
Other graphics were also not explained in the manuscript. Fig. 1 is a copy taken from another paper.
The article needs to be more coherent.
Reviewer 2 Report
Comments and Suggestions for Authors
Thank you for the review invitation for "Motion Capture Technology in Sports Scenarios: A Survey" by Suo et al., which provided a comprehensive overview of the histories of different technologies' advancement and application in motion capture technology in sports scenarios. The manuscript has a great flow, and I have the following minor comments.
Minor comments:
- Figures 2 and 3: I know this figure is for illustration purposes, but its resolution is quite low. Could you please increase it? Also, some of these components are from the public domain; you should include their sources to avoid copyright-related issues.
- For Figure 5, could you please provide more detail on how the scores were given to each of the motion capture technologies? It seems like there is a 5-point system.
Reviewer 3 Report
Comments and Suggestions for Authors
The main purpose of the work was to achieve greater insight into motion capture technology for sports science investigations
I have studied the research in detail. I thank the authors for their efforts, this research is original and relevant in order to optimizing athletes' techniques, prevent injuries by providing detailed feedback on motion data.
The introduction is too short for readers. This section should be improved, I propose to expand the information on application in postural analysis , in fact , inertial sensors provide a cheap and accessible means to efficiently collect and process large amounts of athlete balance data as presented in this paper: doi:10.3390/s23031636.
Regarding methodology, I suggest to explain in more detail the search criteria of articles included and databases used: a more detailed description is needed regarding the included studies.
I suggest adding a table with the repeatability and reliability of the instruments for each applications in order to reinforce the conclusions.
Comments on the Quality of English LanguageModerate editing of English language required
Round 2
Reviewer 1 Report
Comments and Suggestions for Authors
The authors have made some significant changes regarding the consistency of the paper, and some of my remarks have been commented on.
The problem with the figures needs to be resolved correctly. The figure captions in the initial version were ok. Figures should be commented on and explained in the text of the manuscript by pointing out the main steps that characterize the use of a particular motion capture system.
After reading the paper I still do not know what criteria were used to decide that “Robustness” or“Repeatability” or “Reliability” was on “High”, ”Medium” or “Medium to High” levels (Table 2). The authors should explain the method of the performance metrics assessment.
Reviewer 3 Report
Comments and Suggestions for Authors
Congratulations to the researchers. Acceptable for publication if appropriate for the editor and other reviewers.
Comments on the Quality of English Languageminor
Author Response
Thank you for your positive feedback and congratulations on our research. Your endorsement is a significant motivation for our team, and we are grateful for the time and effort you have invested in reviewing our work. We will carefully consider any further suggestions or comments from the editor and other reviewers to further improve the quality of our manuscript. Once again, thank you for your support and encouragement.
